# Attitudes of Caregivers of Children under Five Years Regarding Growth Monitoring and Promotion in Polokwane, Limpopo Province

**DOI:** 10.3390/children10010056

**Published:** 2022-12-27

**Authors:** Mabitsela Hezekiel Mphasha, Matjie Rapetsoa, Nhaviso Mathebula, Kamogelo Makua, Sanele Mazibuko

**Affiliations:** Department of Human Nutrition and Dietetics, University of Limpopo, Sovenga 0727, South Africa

**Keywords:** growth monitoring and promotion, attitudes, caregivers, attendance, children under five

## Abstract

Growth monitoring and promotion (GMP) is critical in tracking child growth to address widespread malnutrition and health status. Attitudes influence behaviour change, including attendance of GMP, and negative attitudes are linked to non-attendance. Moreover, negative attitudes correlate with low socioeconomic position. South Africa is characterized by inequality, which may lead to negative attitudes towards GMP among caregivers with a poor socioeconomic status. Hence, this study seeks to explore the attitudes of caregivers of children under five towards GMP. A qualitative exploratory study design was used. Caregivers of children under five were purposively sampled. Twenty-three participants were interviewed one-on-one, and the data were recorded using voice recorders and field notes. Tesch’s eight steps and inductive, descriptive, and open coding techniques were used to analyse the data. Participants understood the significance of GMP and were confident their children would benefit from it; hence, they attended sessions out of love for their children. The inconsistent availability of GMP services and the behaviour of health workers affected participants’ attitude. Despite these challenges, participants felt good about GMP. Caregivers’ love for their children/grandchildren helped them overcome challenges experienced at the health facilities. Good feelings about GMP boosted caregivers’ attitudes and aided in adherence. An intervention to address element impacting attitudes of caregivers is recommended.

## 1. Introduction

Malnutrition in children is a serious public health issue that harms health and hinders development [1]. Malnutrition is a factor in more than half of all deaths in children under five years because it puts children at risk for developing and dying from infections. There were 144 million stunted, 47 million wasted, 14.3 million severely wasted, and 38.3 million obese children under the age of five years old in the world in 2019 [2,3]. In sub-Saharan Africa, there were 57.5 million stunted, 11.8 million wasted, 3 million severely wasted, and 5.2 million obese children under the age of five years old in 2019 [1]. South Africa (SA) experiences a triple burden of malnutrition—undernutrition, overnutrition, and micronutrient deficiencies [4,5]. Stunted growth is the most prevalent form of malnutrition, and 80% of the world’s stunted children reside in SA [6]. In South Africa, 27% of children under the age of five were stunted, 44% were vitamin deficient, 13% were overweight, and 6% were underweight in 2016 [5]. Malnutrition incidents vary by socioeconomic categories and geographic areas in SA [5]. Growth monitoring and promotion (GMP) is one of the methods that have been used in developing and developed nations to lessen childhood malnutrition [1].

GMP is a nutritional intervention for children under the age of five that involves routinely measuring and plotting their weight on a growth chart at predetermined intervals [7]. A reference group’s growth curves are also compared as part of this process. The findings are utilized to guide caregivers to implement specific actions that will enhance the child’s growth [8,9]. GMP identifies and addresses growth faltering and provides an early warning signal for the implementation of suitable and timely actions before a child’s nutritional status deteriorates into a severe case of malnutrition [1,10]. Immunization, diet, and/or vitamin supplementation, growth monitoring, nutrition education, and counselling are all included in GMP services [11]. GMP has proven to be capable of preserving and improving child-feeding and childcare procedures [12,13]. Attending the GMP sessions allows caregivers to, among other things, speak with healthcare professionals about childcare and receive assurances about the wellbeing of their children [14]. It is important to assess the caregivers’ satisfaction. The level of satisfaction and attitudes towards GMP services are correlated.

According to Vargas-Sánchez, [15] attitude is the way a person perceives and assesses something or someone, as well as the tendency to react favourably or unfavourably to specific concepts, things, people, or circumstances. The positive or negative actions that a person has experienced may influence their attitude. In this context, caregivers’ good attitudes towards GMP services may aid in honoring follow-up appointments. Amid increasing malnutrition among children, it is, therefore, significant to understand the attitudes of the caregivers of children under five years. Muchiri, Gericke, and Rheeder [16] cite attitudes as one of the key factors influencing behaviour change, which, in this context, may influence caregivers to bring their children to GMP sessions. Passive or negative attitudes were found to be linked to low socioeconomic status. [14] Inequality affects people’s viewpoints on things, including involvement in social and health programmes. Researchers have found that those with a lower socioeconomic status experience more stressors, which contribute to a greater number of compulsive behaviours [17,18]. South Africa is characterized by inequality, which may lead to negative attitude towards GMP among caregivers with a poor socioeconomic status, particularly considering that most citizens are poor. This study explores attitudes of caregivers of children under five years regarding growth monitoring and promotion in Polokwane, Limpopo Province. 

## 2. Research Methodology

### 2.1. Research Method and Design

This study was conducted using a qualitative approach, which followed phenomenological exploratory study design. Caregivers of younger children under five years described their attitudes towards GMP, and findings are captured in the Results section.

### 2.2. Study Site

This study was conducted in the rural part of Polokwane Municipality, specifically Sebayeng Clinic. Sebayeng is located within the Solomondale area and falls under Dikgale cluster of Polokwane Municipality. The study site was selected to understand challenges related to non-adherence to GMP sessions. GMP services are offered twice a week at this clinic, mainly by primary healthcare nurses and dietitians. Students from other departments such as nursing and human nutrition and dietetics at the University of Limpopo complete their practicums at this clinic by, among other services, offering GMP services.

### 2.3. Ethical Considerations

The study’s use of human volunteers complied with all required ethical guidelines. The Turfloop Research Ethical Committee (TREC) approved the research and issued clearance certificate number TREC/238/2021:UG. The Limpopo Department of Health (DOH) and the clinic manager gave their permission for the study’s execution. Participants attested to their free will to participate by signing informed consent form. The confidentiality and privacy of the participant’s data were likewise respected.

### 2.4. Sampling and Participants

The target population for this study were caregivers attending GMP services in Sebayeng Clinic. Caregivers in this context refers to mothers and/or legal guardians who assume the primary responsibility for looking after children under the age of five years. As a result, this clinic was chosen because it contains many caregivers of children under five, which made it simpler to find participants who meet the study’s inclusion criteria. Inclusion criteria in this study were only caregivers whose children are enrolled and on the register of GMP programme at Sebayeng Clinic. Twenty-three participants were purposively sampled from the target population of this study. The sample size was established using data saturation. 

### 2.5. Data Collection 

Unstructured interviews were conducted in which participants were interviewed one-on-one. The interviews were recorded using voice recorders, and, to prevent recording unintentional off-the-record remarks, participants were informed when the recorders were on and off. The interviews were conducted by student dietitians doing their community nutrition practicums at the selected clinic. The student dietitians were provided with training on how to conduct one-on-one interviews and ask probing questions from participants. Furthermore, pilot study was conducted in the presence of supervisor to assist in improving the skills of the student dietitians. The participants and results of pilot study were not included in the overall study. The interviews were conducted in Sepedi because the people in the area are Sepedi speakers. The main question that was asked to participants at the opening of interviews was ***‘What is your attitude towards growth monitoring and promotion?’*** An interview guide was used during interviews; however, follow-up questions or probing was completed.

### 2.6. Trustworthiness

Trustworthiness was ensured through credibility, transferability, confirmability, and dependability as described in Table 1.

### 2.7. Data Analysis

Each interview was audio recorded, followed by transcription. Prior to analysis, the researchers had the interviews, which were conducted in Sepedi, translated into English by a language translator. The verbatim transcripts were then independently analysed by each researcher and their supervisors. In a consensus meeting, all researchers and an independent coder came to an understanding on the themes and sub-themes that arose from the independent analysis of the transcripts of the interviews. Direct quotations from participants were taken and recorded to corroborate the conclusions. The eight steps of Tesch’s open coding qualitative data analysis method by Creswell [20] was used to analyse the data. Researchers carefully and repeatedly read all transcripts and developed codes. Topics from codes were grouped together according to similarity, abbreviated, and written next to the appropriate segments of the transcription. Thereafter, themes and sub-themes were developed.

## 3. Results

### 3.1. Demographic Information

The demographic information describes the participants that were interviewed in this study. 

Table 2 shows that all 23 participants were females, 16 were between 18 and 35 years and had a high school education, and 13 were unemployed. In addition, 17 of them were married, and their children were between the ages of 1 year and 30 months old, while 11 of the children were brought by their mothers.

### 3.2. Themes and Sub-Themes

Three themes emerged from this study, which are perceptions on GMP, healthcare services, and adherence to sessions. Themes 1 and 2 had two sub-themes each, while theme 3 had one sub-theme.

#### 3.2.1. Theme 1: Perceptions on GMP

Three essential components of perception are cognition, understanding, and the act of perceiving or comprehending through the senses or intellect. GMP frequently acts as a crucial ongoing link between healthcare workers and a child’s caregivers, by using a growth chart and counselling. The attitudes and believes of laypeople towards GMP are influenced by personal experience. The study’s findings revealed how people view the GMP programme and how well they comprehend GMP. From this theme, the following sub-themes emerged:

#### Sub-theme 1.1: Importance of GMP 

Understanding and appreciating GMP’s worth or significance could help caregivers develop a positive attitude for continuous use of the services. The value of GMP in relation to childcare and health was demonstrated by study participants. The quotes listed below support this sub-theme: 

*“The GMP is vital for child growth and health; it helps identify children with growth faltering for corrective measures and developmental milestones, so that immediate interventions can be initiated”* said **participant 02**.

**Participant 21:** *“I am aware of the significance of bringing the child to the clinic, particularly in order to prevent infections, chicken pox, diarrhea, and vitamin deficiencies.”*

#### Sub-theme 1.2: Feelings towards GMP

Negative feelings towards GMP may lead to skipping regular clinic visits. On the other hand, feeling good might make a person use the services regularly. Participants in this study expressed happiness or feeling good, as seen by the following quotes: 


**Participant 08:**
*“Despite difficulties such as inconsistent availability of GMP services at the clinics, I am ready to keep attending the regular sessions. I’m pleased with this program. It benefits our kids.”*


*“I’ll advise new or young mothers to bring their kids to the GMP sessions. The kids benefit from this. We must put the difficulties we face in the clinic behind us, and continue to bring our children for these services”* said **participant 12.**

#### 3.2.2. Theme 2: Health Services

Healthcare professionals usually deliver GMP services to the caregivers. The availability of services and access to them are essential for adherence. According to the following sub-themes, caregiver attitudes were apparently influenced by the irregular availability of GMP services and the behaviours of the healthcare professionals:

#### Sub-theme 2.1: Inconsistent Availability of GMP Services Impacts Attitudes of Caregivers

The general public’s constant access to and the availability of healthcare services is essential for enhancing the value of GMP services, including attitudes. In contrast, irregular or inconsistent provision of GMP services may result in caregivers not complying, devaluing the programme, and adopting unfavourable attitudes. As evidence, consider the following claims: 

*“We occasionally visit clinics for GMP without assistance on account of unavailability of services, which makes us consider how important the GMP, as promoted, is. Our attitudes and perceptions of the GMP, particularly adherence, are impacted by this”* said **participant 04**.


**Participant 13:**
*"Sometimes we are told that the clinic’s weighing scale is broken or that there are no child immunizations or supplements, so we end up skipping appointments."*


#### Sub-theme 2.2: Behaviour of Healthcare Hroviders and Attitudes of Caregivers 

The healthcare providers who are the drivers or implementers of GMP services to the communities are crucial and may influence caregivers’ adherence to sessions. Additionally, the behaviour of these providers may influence the attitudes of the caregivers on how they perceive GMP. This is supported by the following statement from one of the study’s interviewees:

*“The nurses sometimes speak to us with disrespect, as if we chose for our children not to grow well; the disrespect demotivates us from adhering to sessions. Truly, we turn to develop bad attitudes towards the clinic services including GMP”* said **participant 08**.

#### 3.2.3. Theme 3: Adherence to Sessions

Adherence to the GMP sessions is critical to the health and development of the children, who are usually brought to the healthcare facilities by their caregivers. On the other hand, adherence may be impacted by a number of factors, such as the actions of healthcare workers and the irregular provision of services at the clinics. The following sub-theme shows that caregivers responded that they continue to bring their children to the GMP sessions despite these difficulties or factors out of a sense of love:

#### Sub-theme 3.1: Attendance of Sessions for the Love of Children

Attending the GMP sessions is essential, since it enables early growth changes in the child to be identified. The growth trends are analysed, and the necessary steps are taken. Participants stated that they try to routinely attend the GMP sessions for their children’s sake. The quotations that follow support this: 

*“I love my child and want him to develop into a healthy adult. I go to the regular clinic appointments on time so that I can find out how my child is doing and provide better care”* said **participant 10**.

*“I routinely attend the sessions as scheduled because I want my granddaughter to grow up healthy like my children and free from diseases as well”* said **participant 15.**

## 4. Discussions

The key to influencing behaviour is attitude. When they feel good about the programme or have a positive attitude towards it, the caregivers of children under the age of five are more likely to attend the GMP sessions. Therefore, the purpose of this study was to determine attitudes towards GMP among caregivers of children under five in Polokwane, Limpopo Province. The study showed that participants are aware of the significance of GMP. Various studies found that knowledge is related to attitudes as well as behaviour [21,22]; hence, knowledge emerged during the exploration of attitudes. However, the relationships between the knowledge and attitudes of these caregivers should be determined through a quantitative cross-sectional study, where statistical analysis shall be used to determine associations. Studies have shown a correlation between a knowledge of GMP and higher attendance in the programme, and, conversely, a lack of knowledge was linked to low programme attendance. These researchers showed that caregivers are more willing to participate in the GMP activities on a regular basis when they see them as vital and understand them [23,24]. Knowledge is not the only factor; limited community interaction and inadequate child nutrition counselling, such as from the GMP programme, were found to be the main reasons for low attendance, according to an Ethiopian study that examined reasons for non-attendance [25]. Caregivers desire healthy growth and development, including adequate nutrition, for their children. However, some factors such as availability and affordability may impact feeding. An Australian study reported that low socioeconomic status is linked to a lower intake of healthy food choices. It further proposed the need for policy action to improve the affordability of healthy foods [26]. Participants in this study made references to doing it out of love for their children. Even in the case of the irregular availability of these services and/or unwelcoming behaviour or attitudes of healthcare staff, caregivers may be motivated by love for their own children to adhere to the GMP sessions. Love has the capacity to profoundly alter how people approach their pursuit of health. People who believe they have a health problem or are ill are deemed to be engaging in health-seeking behaviour when they take any activity with the goal of finding a remedy that will work for them [27]. By engaging in appropriate healthcare-seeking behaviour, mothers can either prevent or considerably reduce the likelihood of child mortality brought on by childhood illnesses [28,29,30].

The interviewees in this study made it clear that the inconsistent availability of GMP services and the actions of the healthcare professionals who are the programme’s driving forces have an impact on their perceptions or attitudes regarding this programme. GMP services, such as immunization, supplementation, growth monitoring, nutrition education, and counselling, are carried out by health professionals. According to a cross-sectional survey, caregivers complained that healthcare professionals failed to involve them in growth-monitoring processes and provided generally inadequate GMP counselling [1]. Another study found that after weighing sessions, healthcare professionals did not talk with caregivers about their children’s growth and development and did not ask them to interpret the growth chart [8]. Along with providing a service, the actions of healthcare professionals may also involve how they speak to the caregivers. Hence, primary healthcare facilities should conduct a GMP service-delivery survey to enable users to provide feedback. In addition, it may be significant for clinic managers to address factors that should enhance the attitude of healthcare workers, which include in-service training; this is because their attitude affects the attitude of caregivers. A mixed-method study carried out in Ghana found an association between higher attendance and greater growth, which supports the need to enhance service delivery by strengthening primary healthcare institutions. Additionally, there is a need to raise awareness among caregivers about the value of growth monitoring and promotion for early infant development and encourage engagement through home visits [11]. This will give facilities the chance to reflect and act appropriately to enhance the delivery of these services. Among other things, the Sustainable Development Goals (SDGs) strive to minimize child malnutrition [31,32]. Adherence to the GMP sessions is helpful in the fight against child malnutrition, so it is crucial to make sure that primary healthcare centers have the resources necessary to provide dependable GMP services. In order for caregivers to consistently feel positive about this programme, it is crucial to address variables that may have an impact on their opinions or attitude. All children under the age of five must follow GMP to be healthy. However, the first two years of a child’s life are particularly important [33]. These years have been described as a vital window of opportunity for ensuring that children are appropriately fed to support healthy growth [34]. Children may be more susceptible to malnutrition in the first two years if GMP is not applied properly [35].

Participants in the study also mentioned that they felt good about the programme and looked forward to attending sessions despite the difficulties of the health workers’ behaviour and the irregular availability of GMP. Feeling good about this programme is greatest foundation and positive attitudes. It might encourage participation and adherence of GMP sessions. Feeling good about GMP may be related to participants’ awareness of the significance of this programme and their unwavering love for their children. However, this attitude or feeling good about the GMP programme should be enhanced through the improved service delivery of this programme, including both monitoring and the promotion component of GMP. The study also suggests a programme to encourage the use of GMP services that is motivated by the caregivers’ love for their grandchildren or children.

## 5. Conclusions

According to this study, caregivers are knowledgeable about the significance of GMP and consistently attend sessions because they value, care for, and love their children and grandchildren. The irregular provision of services and the behaviours of the healthcare professionals who are the implementers of this significant programme have an impact on caregivers’ attitudes. It is important to investigate additional variables that might affect caregivers’ attitudes. At the same time, primary healthcare facilities should conduct a GMP service-delivery survey to enable users to provide feedback, for the purpose of improving services. The study also suggests a programme to encourage use of GMP services that is motivated by caregivers’ love for their grandchildren and/or children. Caregivers reported feeling good about this programme, which boosted their attitudes.

**Implications:** There is the need for an intervention to address factors that may have an impact on the attitude of caregivers, to achieve optimal use of the GMP programme.

**Limitations:** This study’s findings demonstrated that the caregivers of children under the age of five were aware of the importance of the GMP programme, but they did not establish a relationship between knowledge and attitude. It also emphasized how caregivers’ attitudes are impacted by healthcare personnel’s attitudes, though it did not evaluate how severe an influence this has. Additionally, this research did not quantify or categorize the participants’ attitudes. This study did not include those who were not enrolled for GMP at Sebayeng Clinic, which could have added a different perspective on attitudes.

## Figures and Tables

**Table 1 children-10-00056-t001:** Measures of trustworthiness.

Strategy	Criterion	Applicability
Credibility	Prolonged engagement	Researchers responsible for data collections were student dietitians conducting community nutrition practicals at Sebayeng Clinic. Their practicals included offering GMP services to caregivers. Therefore, researchers were familiar with and known in the area. In addition, researchers collected data from each participant for about 15–45 minutes to extract more information.
	Member checks	Immediately after the conclusion of each interview, a summary was provided to the participant for confirmation of findings.
Transferability	Data saturation	Participant number 20 added no new data; an extra 3 participants were further sampled for confirmation of data saturation. No new information was added; data collection was, therefore, discontinued.
	Thick description	The study’s setting, sampling procedures, and data collection methods were all clearly explained.
Confirmability	Peer review	Three researchers collected and analysed the data independently, developed themes and sub-themes independently, and met for consensus. Thereafter, they met with supervisors and agreed on themes and sub-themes.
	Reflexivity	The researchers remained neutral during the process of data collections and used probing and reflexivity to acquire more data from participants by using statements such as “*so what you are actually saying is...”*
Dependability	Dense description of research methods	The researchers depended on clear and detailed description of research methods, including the use of participants’ quotations in presenting results.
	Audit trail	Researchers who collected the data described to supervisors how the data were gathered and analysed. Themes and sub-themes were agreed upon. Data to be kept in a lockable file for the ensuing five years.
	Reliance on data collection tools, and supervisors	Researchers relied on voice recorder to capture interviews and experiences of supervisors.

Source: Ramathuba and Ndou [19].

**Table 2 children-10-00056-t002:** Socio-demographic profile of participants.

Socio-Demographic Profile	Category	N = 23
Gender	Male	0
	Female	23
Age	18–35	16
	36 or above	7
Marital status	Married	17
	Single	6
Employment	Employed	10
	Unemployed	13
Education	High school education or less	16
	Tertiary education	7
Child’s age	1–30 months	17
	31–72 months	6
Relationship to child	Mother	11
	Grandmother	10
	Sibling of mother	2

## Data Availability

This article depends on the data gathered from the caregivers of children between 6–24 months in Sebayeng Clinic in Polokwane, Limpopo Province. The data generated or analysed during the current study are not openly accessible due to planned publication. However, they can be requested from the corresponding author by email: pitso85@gmail.com.

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
