# Peer review of "Attitudes of Caregivers of Children under Five Years Regarding Growth Monitoring and Promotion in Polokwane, Limpopo Province"

_children, 2022, doi:10.3390/children10010056_

Round 1

Reviewer 1 Report

Growth Monitoring and Promotion (GMP) is an important concept introduced to improve child nutritional outcomes, health and reduce child deaths. GMP emphasises linking the results of monitoring with follow‐up promotional actions and its success in LMICs with limited resources depends heavily on the attitude and utility of key beneficiaries besides supply-side challenges. In this context, the paper appears to be interesting in adding evidence towards improving GMP in LMICs.

While much research is needed in the current area, I wonder if the current work manages to provide significant and robust evidence. The comments below are made with the intention to help and I hope they are seen in that light.

 Abstract: The abstract need to be clear and self-explanatory. For e.g., in line 13, it was not clear on what the study seeks to explore specifically (? GMP). Overall, the abstract need to have more focus on results by condensing the methods. A line on implication/conclusion should be added to the abstract for the benefit of readers.

Introduction: English language and grammar need revisions at places throughout the manuscript. Line 16 (23 participants were voice recorders); line 65 (passive); line 83 (dietititians).

The authors need to build a strong case for their study by expanding on the need for understanding/exploring the attitudes of the caregiver towards GMP. The authors need to clarify how inequality leads to negative attitudes. Line 70 & 71 appears redundant after citing the objectives.

Methods: The authors need to be consistent in their objectives and methodology, while abstract & introduction cites objective as “explores attitudes of caregivers of children under five years regarding GMP”, the results capture data on “factors affecting non-adherence to GMP sessions among caregivers of under-five children” which is not coherent and appropriate.

Sampling : Line 98, it was not clear what the authors meant by “Population size of this study was estimated at 300 caregivers”.

The authors have used the term “purposive sampling” in the abstract, however, they have used the term “conveniently sampled” in the sampling section (line 105). Both sampling strategies are quite different and the authors need to be consistent in using valid and appropriate terminology.

The study did not include caregivers who were not enrolled for GMP. Missing out such negative or deviant cases would have impact on the credibility and maximum diversity sampling. The authors need to discuss this in the limitation section.

The authors need to provide information on the training and experience of student dietitians in conducting one-on-one interviews.

Results : The table on the socio-demographic profile can include the relationship of caregiver with the child.

In line 145-147, the authors have informed that three themes have emerged, but the result section and table 4 does not provide any information on the 2nd and 3rd theme. Also, the theme “Attitudes towards GMP” were used interchangeably with “Knowledge related to GMP” which is not correct.

Some of the subthemes in Table 4 appear non-related to the theme1.

Overall, the result section is very terse, abrupt, and incomplete leading to poor clarity and inadequacy in addressing the proposed research question.

Discussion: with incomplete results, the discussion appears disconnected in linking the available evidence to the context.

Line 206; the authors should avoid using the term “discovered” in this technical paper.

More importantly, the authors did not discuss the limitations and implications of the study findings.

Overall, the authors need to provide a clear picture of the study results and expand on the discussion section with relevant evidence by linking results with the implications. Thus, the manuscript requires extensive revision to answer the proposed research questions in a more robust manner for better clarity and validity.   

Author Response

Reviewers comment

Implementation

Page number

The abstract need to be clear and self-explanatory. For e.g., in line 13, it was not clear on what the study seeks to explore specifically (? GMP).

The aim of the study has been clearly put

Page 2, line 12

Overall, the abstract need to have more focus on results by condensing the methods. A line on implication/conclusion should be added to the abstract for the benefit of readers.

Methodology condensed and results expanded as suggested.

Page 2, line 12 – 22

English language and grammar need revisions at places throughout the manuscript. Line 16 (23 participants were voice recorders); line 65 (passive); line 83 (dietititians).

Grammer corrected

Whole manuscript

The authors need to build a strong case for their study by expanding on the need for understanding/exploring the attitudes of the caregiver towards GMP.

Done as recommended

Page 2, line 55-59

The authors need to clarify how inequality leads to negative attitudes. Line 70 & 71 appears redundant after citing the objectives.

Clarified

Page 2, line 61-63

The authors need to be consistent in their objectives and methodology, while abstract & introduction cites objective as “explores attitudes of caregivers of children under five years regarding GMP”, the results capture data on “factors affecting non-adherence to GMP sessions among caregivers of under-five children” which is not coherent and appropriate.

Exploring attitudes maintained as objective

Page 2, line 12

Page 3, line 66-67

Page 3, line 73

Sampling : Line 98, it was not clear what the authors meant by “Population size of this study was estimated at 300 caregivers”.

It was an error and deleted

The authors have used the term “purposive sampling” in the abstract, however, they have used the term “conveniently sampled” in the sampling section (line 105). Both sampling strategies are quite different and the authors need to be consistent in using valid and appropriate terminology.

Purposive sampling maintained

Page 2, line 13

Page 3, line 99

The authors need to provide information on the training and experience of student dietitians in conducting one-on-one interviews.

Provided

Page 3, line 103-106

The table on the socio-demographic profile can include the relationship of caregiver with the child.

Included

Page 4, line 127

In line 145-147, the authors have informed that three themes have emerged, but the result section and table 4 does not provide any information on the 2nd and 3rd theme. Also, the theme “Attitudes towards GMP” were used interchangeably with “Knowledge related to GMP” which is not correct.

It was an error and revised accurately

Page 4, line 132-133

Overall, the result section is very terse, abrupt, and incomplete leading to poor clarity and inadequacy in addressing the proposed research question.

Discussion: with incomplete results, the discussion appears disconnected in linking the available evidence to the context.

Revised accurately

Page 5 – 6, line 135 – 248

More importantly, the authors did not discuss the limitations and implications of the study findings.

Included

Page 6-7, line 251-265

Overall, the authors need to provide a clear picture of the study results and expand on the discussion section with relevant evidence by linking results with the implications. Thus, the manuscript requires extensive revision to answer the proposed research questions in a more robust manner for better clarity and validity.  

The discussions has been expanded

Page 6, line 211-216

Page 7, line 241-245

Reviewer 2 Report

I donțt think that this paper can arouse the interest of the readers. It is not a very solid scientific work. It needs to be improved. There are not so much statistical data that could sustain the usefulness of your work. It's more like a short analysis of your daily work.

Author Response

The paper has been revised and improved accordingly

Reviewer 3 Report

This qualitative research paper about the attitude of caregivers is not very scientific, but is a start for more information about the problems in monitoring and promotion of growth in under 5 year old children.

Their is a start of research in poor and difficult situations such as language (sepedi) in the interviews.

Table 1 is not included in text and spelling (dietititions, in detailed etc.) anadn other spelling faults.

Author Response

Grammar and language has been improved

Round 2

Reviewer 1 Report

Though the manuscript has been improved, I can see extensive revisions at places in the revised manuscript which raises some concern. The comments below are made with the intention to help and I hope they are seen in that light.

1     1.  English language and grammar need revisions at places throughout the manuscript for example; Line 15 etc.,

2     2. After the stated aims, it's unclear how the lines 67–68 is relevant.

3     3. The sentence in line-71 appears incomplete

4     4. In line-92, the authors have mentioned Inclusion criteria in this study were only caregivers whose children regularly attends GMP sessions at Sebayeng clinic”; The authors need to mention what they meant by regular and clarify how such regular participants can provide narratives on the non-adherence/skipping appointments to GMP.

5    5. Table 2 was missing in the revised manuscript

      6. Results: This is a major concern; The authors state that they have arrived at only one theme which is highly unlikely. Also, the theme itself is the topic of the study. The authors need to revisit their qualitative analysis, refine, and categorise their themes and sub-themes further for meaningful interpretation and robustness.

7   7. The study did not include caregivers who were not enrolled for GMP. Missing out such negative or deviant cases would have impact on the credibility and maximum diversity sampling. The authors need to discuss this in the limitation section.

Author Response

The whole manuscript has been proofread and edited to correct gramma

After the stated aims, it's unclear how the lines 67–68 is relevant. THIS HAS BEEN DELETED

The sentence in line-71 appears incomplete. IT HAS BEEN COMPLETED 

In line-92, the authors have mentioned “Inclusion criteria in this study were only caregivers whose children regularly attends GMP sessions at Sebayeng clinic”; The authors need to mention what they meant by regular and clarify how such regular participants can provide narratives on the non-adherence/skipping appointments to GMP. IT HAS BEEN CLARIFIED

Table 2 was missing in the revised manuscript. IT WAS AN ERROR IN NAMING, WHICH HAS BEEN ADDRESSED

Results: This is a major concern; The authors state that they have arrived at only one theme which is highly unlikely. Also, the theme itself is the topic of the study. The authors need to revisit their qualitative analysis, refine, and categorise their themes and sub-themes further for meaningful interpretation and robustness. IT HAS BEEN REVIEWED PROPERLY AND 3 THEMES INCLUDED

The study did not include caregivers who were not enrolled for GMP. Missing out such negative or deviant cases would have impact on the credibility and maximum diversity sampling. The authors need to discuss this in the limitation section. IT HAS BEEN ADDRESSED IN THE LIMITATIONS AS SUGGESTED

Reviewer 2 Report

There is some improvement in the paper you sent for reviewing.

I still can't find strong sttatistical data to support this paper.

In my belief this paper still needs to be improved, following the instructions I sent last time.

Author Response

I still can't find strong statistical data to support this paper. THIS WAS A QUALITATIVE STUDY WHICH DOES NOT NECESSARILY PRESENT STATISTICAL ANALYSIS. HOWEVER, THE DISCUSSIONS OF THE FINDINGS ARE SUPPORTED BY VARIOUS QUALITATIVE, QUANTITATIVE AND MIXED METHOD STUDY. FURTHERMORE, IT HAS BEEN NOTED AS LIMITATIONS THAT THE STUDY DID NOT ASSESS CORRELATIONS WHICH WILL REQUIRE STATISTICAL ANALYSIS.

Round 3

Reviewer 1 Report

The authors have addressed most of the concerns and the current revision appears more scientific and has  brought better clarity.

In line 120, "source" can be deleted.

As the authors have removed Table 2 describing Tesch's open coding qualitative data analysis method (in the first version), it would  benefit the readers if the authors can expand a line or two describing their qualitative analysis on arriving at themes and sub-themes.

Author Response

The 'source' was delete

As the authors have removed Table 2 describing  Tesch's open coding qualitative data analysis method (in the first version), it would benefit the readers if the authors can expand a line or two describing their qualitative analysis on arriving at themes and sub-themes. IT WAS ADDED ON PAGE 4, LINE 122-124

Reviewer 2 Report

Please check again the spelling. I must admit that you've done a lot of improvement.

Author Response

Further spelling was checked and improved